Diagnostic and prognostic implications of growth differentiation factor 15 in heart failure with preserved ejection fraction: a systematic review and meta-analysis

Dakota Iwan 1 2
http://orcid.org/0000-0001-8601-8209 Wijayanto Matthew Aldo 1 3 matthewaldo1810@gmail.com
http://orcid.org/0000-0002-5450-4568 Nugrahani Annisa Salsabilla Dwi 4
Sunjaya Angela Felicia 1
http://orcid.org/0000-0003-3799-8102 Rachmayanti Shela 1
Indah Enny Yuliana 1
Naomi Natasya 1
Huang Wilbert 1
http://orcid.org/0009-0009-4105-8739 Tristan Christopher Daniel 3
Fauziah Mira 1 2
Muliawan Hary Sakti 2 5
http://orcid.org/0000-0003-3998-1590 Siswanto Bambang Budi 1 2
1 National Cardiovascular Center Harapan Kita , West Jakarta, Jakarta , Indonesia
2 Department of Cardiology and Vascular Medicine, Faculty of Medicine, Universitas Indonesia , Depok, West Java , Indonesia
3 Faculty of Medicine, Universitas Sebelas Maret , Surakarta, Central Java , Indonesia
4 Medical Program, Faculty of Medicine, Universitas Airlangga , Surabaya, East Java , Indonesia
5 Department of Cardiology and Vascular Medicine, Universitas Indonesia Hospital , Depok, West Java , Indonesia
Connor Mark
Electronic publication date: 2025 Oct 28
Publication date: 2025
Volume: 13
Electronic Location ID: e20168
Received 2025 Apr 16; Accepted 2025 Sep 11
Copyright: © 2025 Dakota et al.
Copyright year: 2025
Copyright holder: Dakota et al.
License: This is an open access article distributed under the terms of the Creative Commons Attribution License, which permits unrestricted use, distribution, reproduction and adaptation in any medium and for any purpose provided that it is properly attributed. For attribution, the original author(s), title, publication source (PeerJ) and either DOI or URL of the article must be cited.
License URL: https://creativecommons.org/licenses/by/4.0/

Keywords: Growth differentiation factor-15, Heart failure with preserved ejection fraction, Biomarker, Systematic review, Meta-analysis

Funding: The authors received no funding for this work.

==============================
Background

Growth differentiation factor-15 (GDF-15), an emerging biomarker associated with chronic inflammation and oxidative stress, shows potential diagnostic and prognostic significance for heart failure with preserved ejection fraction (HFpEF). This study aimed to assess the diagnostic and prognostic value of GDF-15 in HFpEF.

Methods

Three databases (PubMed, Scopus, and ScienceDirect) were used to search for relevant literature published before August 6, 2024. Quality assessment was conducted using the Newcastle-Ottawa scale and its adaptation for cross-sectional studies. Statistical analysis was performed using RStudio version 4.4.1. All meta-analyses employed a random-effects model. Sensitivity analysis was conducted using the leave-one-out technique to evaluate the influence of individual studies on pooled estimates. This study protocol was registered in PROSPERO (CRD42024569609).

Results

A total of 5,696 HFpEF patients were identified from 28,193 individuals across 12 observational studies. GDF-15 levels were consistently elevated in HFpEF patients, with a pooled mean difference (MD) of 647.60 pg/mL (95% CI [148.43–1,146.77]; p = 0.01). Sensitivity analysis confirmed the robustness of this finding, with a slightly higher MD observed when studies involving HFpEF patients with atrial fibrillation were excluded. Qualitative analysis suggested that the overall diagnostic performance of GDF-15 in HFpEF is slightly superior to conventional biomarkers. GDF-15 showed a pooled area under the curve (AUC) of 0.82 (95% CI [0.72–0.91]), indicating good diagnostic accuracy. Additionally, GDF-15 was associated with increased risk of all-cause mortality and heart failure hospitalisation, with pooled hazard ratios (HR) of 1.46 (95% CI [1.30–1.62]; p < 0.01) and 1.76 (95% CI [1.30–2.38]; p < 0.01), respectively.

Conclusion

GDF-15 demonstrates significant diagnostic and prognostic potential for HFpEF. Elevated GDF-15 levels are associated with increased risk of all-cause mortality and heart failure hospitalisation.

Introduction

Heart failure with preserved ejection fraction (HFpEF) has become a significant public health concern, affecting approximately 50% of the global heart failure (HF) population (Borlaug & Paulus, 2011; Borlaug et al., 2023). The 5-year mortality and rehospitalisation rates for this condition are comparable to those observed in HF patients with reduced ejection fraction (HFrEF) (Borlaug et al., 2023). However, the underlying cause of HFpEF is not fully understood, leading to challenges in accurately diagnosing the condition and selecting a suitable treatment (Borlaug, 2020; Nagueh, 2021; Mishra & Kass, 2021). HFpEF remains a challenging condition to diagnose, with conventional biomarkers such as natriuretic peptide often failing to meet diagnostic thresholds, particularly in those with obesity-related HFpEF, despite elevated left ventricular (LV) filling pressures at rest or during exercise (Santhanakrishnan et al., 2012; Borlaug et al., 2023). This limitation highlights the need for novel biomarkers that better reflect the pathophysiology of HFpEF.

Growth differentiation factor-15 (GDF-15), a novel biomarker that has the potential to be a valuable diagnostic and prognostic tool in HF, has been associated with chronic inflammation and oxidative stress (di Candia et al., 2021; Morfino et al., 2022). GDF-15 appears to more effectively represent this ongoing low-grade inflammatory condition than other conventional biomarkers (Chan et al., 2016), as HFpEF is a phenotype that is primarily associated with a chronic inflammatory state (Shah et al., 2016). Prior meta-analysis demonstrated that GDF-15 is a promising biomarker for diagnosing HFpEF, with a pooled area under the curve (AUC) of 0.88, indicating strong diagnostic accuracy (Wang, Gu & Guo, 2023). However, the study was limited by a small sample size (1,550 patients), and it did not assess the prognostic value of GDF-15 in HFpEF.

Examining a biomarker comprehensively provides a strong means of contextualising the biomarker and enhancing the knowledge of disease-related pathways, as the utilisation of this approach can optimise clinical decision-making, thereby improving overall patient outcomes. Therefore, this systematic review and meta-analysis aimed to systematically analyse, from published literature, the diagnostic and prognostic value of GDF-15 in HFpEF.

Materials and Methods

This systematic review and meta-analysis were conducted in accordance with the Preferred Reporting Items for Systematic Reviews and Meta-analyses (PRISMA) reporting guideline (Page et al., 2021). The present study was registered in the international prospective register of systematic reviews (PROSPERO) under the protocol number CRD42024569609.

Study eligibility

We included studies that fulfilled the following eligibility requirements: (i) observational studies (cohort, case-control, cross-sectional, or post hoc randomised controlled trial (RCT)); (ii) studies assessing GDF-15 in HFpEF populations, according to the latest 2021 European Society of Cardiology definition of HFpEF. Review articles, letters, commentaries, case reports, and case series were excluded from the study. Only articles written and published in English were included, with no restrictions in regard to dates. HFpEF diagnosis criteria were defined by using the standards of the European Society of Cardiology 2021 HFpEF diagnosis criteria: (i) symptoms and signs of HF; (ii) a left ventricular ejection fraction (LVEF) ≥50%; and (iii) objective evidence of cardiac anatomical and/or functional abnormalities consistent with the presence of LV diastolic dysfunction/raised LV filling pressures, including elevated natriuretic peptides (McDonagh et al., 2021).

Search strategy and data extraction

A systematic literature search was performed on Scopus, PubMed, and ScienceDirect databases on August 6, 2024, using search terms (“Heart Failure with Preserved Ejection Fraction” OR “Diastolic Heart Failure” OR “Heart Failure, Normal Ejection Fraction”) AND (“Growth Differentiation Factor 15” OR “GDF#15” OR “Macrophage Inhibitory Cytokine 1”). To identify eligible literature, two independent authors (SR and AFS) conducted abstract and full-text screenings using the Rayyan software (Ouzzani et al., 2016). Disagreements were resolved by consensus with the third reviewer (MAW). All identified articles were assessed using the inclusion and exclusion criteria.

The following data were extracted from each included study: (i) publication details (country, year of publication, and first author’s last name); (ii) demographic characteristics (sample size, mean age, and mean ± standard deviation (SD) or median interquartile range (IQR) values of LVEF); (iii) study details (study design, serum GDF-15 data, including units, measurement methods, and assay kits); (iv) diagnostic analysis data (control group definition, sample size, comparison of GDF-15 and N-terminal pro-B-type natriuretic peptide (NT-proBNP) diagnostic performance, mean ± SD or median (IQR) values, optimal cut-off values, AUC for the receiver operating characteristic (ROC) curve, sensitivity, and specificity); and (v) prognostic meta-analysis data (follow-up duration, clinical outcomes, unadjusted and multivariable-adjusted effect size with the 95% confidence intervals (CI), and the adjustment variables).

Studies were classified as diagnostic if they evaluated the ability of GDF-15 to differentiate HFpEF from controls through metrics such as AUC. Furthermore, studies that reported GDF-15 levels specifically in HFpEF and control groups were classified as diagnostic. Studies were classified as prognostic if they examined the relationship between GDF-15 levels and associated outcomes in patients with HFpEF.

Quality assessment

Using the Newcastle-Ottawa Scale (NOS) for cohort or case-control studies and NOS adapted for cross-sectional studies, two authors (NN and EYI) independently evaluated the risk of bias in the studies included in the analysis. A score was assigned to each investigation on a scale of zero to nine for NOS and zero to ten for NOS adapted for cross-sectional studies. A research study was deemed to be of high quality if it received a score of 7 or higher. Any discrepancies were resolved through consensus with the third reviewer (BBS).

Statistical analysis

All statistical analyses were performed using R-studio version 4.4.1 (PositPBC, Boston, MA, USA). Numerical data were calculated as the mean difference (MD) between the control and HFpEF groups. Data reported in the median is converted into the mean for the meta-analysis; an online converter was utilised (https://ma-accelerator.com). The diagnostic value (pooled AUC) and prognostic value (all-cause mortality and HF hospitalisation) of GDF-15 in HFpEF were assessed using a random-effects model with the restricted maximum likelihood (REML) estimator with t-distribution confidence interval, employing the generic inverse variance method via the “metagen” function from the “meta” package in R. For the prognostic analysis, we only included studies reporting adjusted estimates. Hazard ratio (HR) from prognostic outcomes was transformed into its natural logarithmic form (ln[HR]) before back-transformation for interpretation. Statistical significance was defined as a p-value of less than 0.05. Heterogeneity was assessed using I2, with values of <25%, 25–50%, and >50% represented low, moderate, and high heterogeneity, respectively. Sensitivity analysis was performed using the leave-one-out technique to evaluate the influence of individual studies on the overall estimate.

Results

Literature search results and quality assessment

The initial search identified 328 records, with 280 remaining after duplicate removal. A total of 17 reports underwent full-text review, but one was not retrieved, and four were excluded due to irrelevant outcomes or duplicate populations. Ultimately, 12 studies were included in the systematic review and meta-analysis (Fig. 1). All but one of the studies were assessed as being of good quality, as summarised in Table S1.

Figure 1 PRISMA flow diagram for study selection.

Characteristics of included studies

Table 1 presents the baseline characteristics of the included studies (Baessler et al., 2012; Santhanakrishnan et al., 2012; Izumiya et al., 2014; Sinning et al., 2017; Jirak et al., 2020; Kanagala et al., 2020; Mendez Fernandez et al., 2020; Mitic et al., 2020; Aulin et al., 2022; Oyama et al., 2023; Yin et al., 2023; Lyu et al., 2024). This systematic review and meta-analysis includes 5,696 HFpEF patients with a variety of study designs, primarily prospective cohorts, conducted across multiple countries, including the United Kingdom, Japan, Germany, China, and multinational settings. From the included studies, the sample sizes varied, ranging from 112 to 11,818 participants, with the number of HFpEF patients ranging from 18 to 2,520. The mean age of HFpEF patients was generally in the elderly range, from 50.3 to 73 years, reflecting the typical demographic affected by HFpEF. The LVEF ranged from 53.6% to 64.0% among HFpEF patients. The studies used different sources for GDF-15 assay kits, such as Biovendor, R&D Systems, Millipore, and Roche Diagnostics, and employed various detection methods, including enzyme-linked immunosorbent assays, Luminex bead-based multiplex assays, and electrochemiluminescent assays. GDF-15 levels among HFpEF patients varied widely, with mean values ranging from 695 to 4,308 pg/mL.

Table 1 Baseline characteristics of the included studies.

Author (Year)	Country	Study design	Sample size
(n)	Specific population	HFpEF patients (n)	Mean Age HFpEF (years)	LVEF of HFpEF patients (%)	Source of GDF-15 Assay Kit	GDF-15 detection method	GDF-15 of HFpEF patients (pg/mL)	Purpose (Diagnostic/Prognostic/Both)	
Baessler et al. (2012)	Germany	Prospective cohort, single-centre	258	Obese	88	50.3 ± 7.3	64.0 ± 9.0	Quantikine®; R&D Systems Europe, Wiesbaden, Germany	Enzyme-linked immunosorbent assay	695.67 ± 324.09	Diagnostic	
Santhanakrishnan et al. (2012)	Singapore	Prospective cohort, multi-centre	151	n/a	50	69.0 ± 12.0	60.0 ± 7.0	Quantikine®; R&D Systems, Inc. Minneapolis, Minnesota, USA	Sandwich enzyme immunoassay	2,708.43 ± 2,336.99	Diagnostic	
Izumiya et al. (2014)	Japan	Prospective cohort, single-centre	149	n/a	73	70.1 ± 11.0	61.7 ± 5.7	Biovendor, Asheville, North Carolina, USA	Enzyme-linked immunosorbent assay	4,308.77 ± 1,483.00	Prognostic	
Sinning et al. (2017)	Germany	Prospective cohort, population-based	5,000	n/a	70	67.0 ± 7.57	64.10 ± 8.02	n/a	Immunoluminometric assay (ILMA)	1,382.70 ± 662.56	Both	
Kanagala et al. (2020)	United Kingdom	Prospective cohort, single-centre	234	n/a	140	73.0 ± 9.0	56.0 ± 5.0	Bristol Myers Squibb, Ewing Township, New Jersey, USA	Luminex bead-based multiplex assay	2,459.67 ± 1,536.05	Diagnostic	
Jirak et al. (2020)	Germany	Retrospective single-centre	252	n/a	18	70.9 ± 6.5	59.7 ± 9.8	R&D Systems, Minneapolis, Minnesota, USA	Enzyme-linked immunosorbent assay	837.97 ± 313.31	Diagnostic	
Mitic et al. (2020)	Serbia	Cross-sectional, single-centre	112	n/a	26	63.8 ± 9.1	53.6 ± 3.7	Quantikine®; R&D Systems, Inc. Minneapolis, Minnesota, USA	Sandwich enzyme immunoassay	1,493.10 ± 421.40	Diagnostic	
Mendez Fernandez et al. (2020)	Spain	Prospective cohort, single-centre	311	n/a	221	73.0 ± 12.0	64.0 ± 9.0	Roche Diagnostics, Basel, Switzerland	Electrochemiluminescence immunoassays	2,897.67 ± 1,851.35	Prognostic	
Aulin et al. (2022)	Multinational	Post hoc RCT, multi-centre	11,818	AF	2,520	68.0 ± 9.64	56.0 ± 8.90	Elecsys pre-commercial assay kit P03 from Roche Diagnostics	Electrochemiluminescence immunoassays	1,490.00 ± 844.49	Diagnostic	
Yin et al. (2023)	China	Prospective cohort, multi-centre	380	New HF or chronic decompensated HF with preserved EF	380	70.67 ± 11.16	59.0 ± 8.93	Millipore, Billerica, Massachusetts, USA	Luminex bead-based multiplex assay	3,490.33 ± 2,191.53	Prognostic	
Oyama et al. (2023)	Multinational	Post hoc RCT, multi-centre	8,705	AF	2,016	70.67 ± 9.64	n/a	Roche Diagnostics, Indianapolis, Indiana, USA	Automated electrochemiluminescent sandwich immunoassay	1,686.00 ± 925.85	Both	
Lyu et al. (2024)	China	Prospective cohort, single-centre	823	CHD	322	73.67 ± 12.66	56.67 ± 5.21	R&D Systems, Minneapolis, Minnesota, USA	Enzyme-linked immunosorbent assay	3,281.33 ± 2,376.38	Both	
Note:

AF, atrial fibrillation; CHD, coronary heart disease; EF, ejection fraction; HF, heart failure; HFpEF, heart failure with preserved ejection fraction; LVEF, left ventricular ejection fraction; n/a, not available; RCT, randomised controlled trial; USA, United States of America.

Diagnostic performance of GDF-15 in distinguishing HFpEF from controls with conventional biomarkers

GDF-15 levels were elevated in HFpEF patients compared to control groups across all studies analysed, with the exception of a study by Oyama et al. (2023) that compared the HFpEF group with a control group of individuals with atrial fibrillation (AF) and no HF (median GDF-15: 1,599.0 vs. 1,674.0 pg/mL). Within the obese population with HFpEF, the levels of GDF-15 were markedly higher than in the obese population with normal LV function (median GDF-15 662 vs. 451 pg/mL) (Baessler et al., 2012) (Tables S2, S3).

Figure 2A depicted that GDF-15 levels were significantly elevated in HFpEF compared to a control group of the non-HF population (MD 647.60 pg/mL; 95% CI [148.43–1,146.77]; p = 0.01; I2 98%). This result was deemed robust with a leave-one-out sensitivity analysis (Fig. 2B), as there was no change in the p-value when omitting any study. However, the MD was slightly higher when studies by Oyama et al. (2023) or Aulin et al. (2022) were excluded, indicating that these two studies contributed to narrowing the overall difference in the meta-analysis. Additionally, the heterogeneity remained unchanged, indicating that the observed variability across studies was not driven by the exclusion of any single study.

Figure 2 Meta-analysis of mean difference in GDF-15 levels among HFpEF vs. controls.

(A) Forest plot of meta-analysis. (B) Sensitivity analysis using the leave-one-out method (Baessler et al., 2012; Santhanakrishnan et al., 2012; Sinning et al., 2017; Kanagala et al., 2020; Jirak et al., 2020; Mitic et al., 2020; Aulin et al., 2022; Oyama et al., 2023).

Four studies estimated the diagnostic performance of GDF-15 in identifying HFpEF from controls (non-HF population (Sinning et al., 2017), obese with normal LV function (Baessler et al., 2012), non-CAD and non-HF population (Santhanakrishnan et al., 2012), and population without HFpEF, dilated cardiomyopathy, and ischaemic cardiomyopathy (Jirak et al., 2020)). From the qualitative analysis, the overall diagnostic performance of GDF-15 in HFpEF appears to be slightly superior compared to NT-proBNP, as a conventional biomarker for HFpEF (Table S4). While the combination of GDF-15 and NT-proBNP resulted in a higher AUC (0.956; 95% CI [0.919–0.994]), this combined biomarker strategy was not statistically different from using GDF-15 or NT-proBNP alone (p = 0.31; p = 0.33, respectively) (Santhanakrishnan et al., 2012). Figure 3A depicted the pooled AUC of GDF-15 for HFpEF (0.82; 95% CI [0.72–0.91]; I2 84%), indicating an overall good diagnostic accuracy. Sensitivity analysis reveals a robustness in the result with unchanged statistical significance and direction of effect. However, there was a reduction in heterogeneity when omitting the study by Santhanakrishnan et al. (2012), indicating that this study was a major contributor to the observed heterogeneity (Fig. 3B).

Figure 3 Meta-analysis of pooled AUC of GDF-15 levels in HFpEF population.

(A) Forest plot of meta-analysis. (B) Sensitivity analysis using the leave-one-out method (Baessler et al., 2012; Santhanakrishnan et al., 2012; Sinning et al., 2017; Jirak et al., 2020).

GDF-15 and prognostic outcomes in patients with HFpEF

A total of six studies evaluated the prognostic implications of serum GDF-15 in relation to all-cause mortality, adverse cardiovascular events, and HF hospitalisation with a mean follow-up duration ranging from 12 to 112 months (Tables S5–S7) (Izumiya et al., 2014; Sinning et al., 2017; Mendez Fernandez et al., 2020; Oyama et al., 2023; Yin et al., 2023; Lyu et al., 2024). A prospective cohort reported that GDF-15 concentrations independently predict all-cause mortality in the Spanish population based on unadjusted multivariate analysis (Mendez Fernandez et al., 2020). Additionally, elevated serum GDF-15 levels were significantly associated with an increased risk of adverse cardiovascular events in HFpEF patients, underscoring its prognostic value in this population (Izumiya et al., 2014; Oyama et al., 2023).

From three studies included in the meta-analysis, elevated GDF-15 was significantly increased the hazard of all-cause mortality with a pooled HR of 1.46 (95% CI [1.30–1.62], p < 0.01; I2 0%) (Fig. 4A). The resulted did not differ when sensitivity analysis was performed (Fig. 4B). Two studies also evaluated the effect size of GDF-15 and HF hospitalisation (Yin et al., 2023; Lyu et al., 2024), with a pooled HR of 1.76 (95% CI [1.30–2.38]; p < 0.01; I2 0%) (Fig. 4C), indicating a significant association between elevated GDF-15 levels and increased risk of HF hospitalisation. The findings indicate that GDF-15 may serve as a valuable prognostic biomarker, especially for assessing all-cause mortality and HF hospitalisation risk in clinical settings.

Figure 4 Meta-analysis of prognostic value of GDF-15 in HFpEF population.

(A) Pooled HR of GDF-15 for all-cause mortality. (B) Sensitivity analysis of all-cause mortality using the leave-one-out method. (C) Pooled HR of GDF-15 for HF hospitalisation (Sinning et al., 2017; Yin et al., 2023; Lyu et al., 2024).

Discussion

In this systematic review and meta-analysis, a total of 5,696 HFpEF patients were included from 28,193 participants. GDF-15 levels were significantly higher in the HFpEF patients compared to the control group of the non-HF population. GDF-15 has a good diagnostic accuracy in diagnosing HFpEF and appears to be slightly superior compared to conventional biomarkers. GDF-15 also demonstrates significant prognostic potential in HFpEF, as higher levels of GDF-15 are associated with an increased risk of all-cause mortality and HF hospitalisation.

HFpEF is a complex disease with interrelated pathophysiological mechanisms (Logeart, 2024). The condition’s prevalence has surged to the point of being considered an epidemic, with recent research highlighting that HFpEF encompasses a range of distinct clinical phenotypes. These phenotypes, although grouped under the same definition, exhibit diverse pathophysiological characteristics and mechanisms (Stoicescu et al., 2024). Over the last decade, several groups of investigators were able to obtain myocardial tissue from HFpEF patients, revealing specific alterations in myocardial structure, function, and intramyocardial signaling, which were relevant to the concentric LV remodeling and diastolic LV dysfunction characteristically observed in patients with HFpEF (van Heerebeek & Paulus, 2016). These alterations include increased myocardial fibrosis, heightened oxidative stress, endothelial dysfunction, and chronic low-grade inflammation—all of which are associated with disease progression in HFpEF. HFpEF is additionally marked by a sustained elevation in inflammatory biomarkers, including GDF-15. This cytokine is elevated due to systemic inflammation and oxidative stress, both of which drive myocardial remodeling and endothelial dysfunction in HFpEF. Inflammation may be a primary factor in the development and advancement of HFpEF and its multiple related comorbidities, and GDF-15 levels often reflect this ongoing cellular stress and inflammatory burden of HFpEF (Lewis et al., 2022; Pugliese et al., 2023).

Although NT-proBNP is a valuable biomarker in HF, it may have limitations in HFpEF due to confounding effects from prevalent comorbidities such as AF, obesity, and renal impairment. NT-proBNP levels in HFpEF patients can be deceptive. For example, AF may elevate NT-proBNP in a manner that is disproportionate to the severity of HF, whereas obesity has a tendency to lower NT-proBNP levels in patients with significant HFpEF-related pathology. In addition, renal impairment complicates the interpretation of NT-proBNP, as reduced clearance can result in artificially elevated levels, further complicating the clinical picture (Januzzi & Myhre, 2020). Our study showed that GDF-15 appears to be slightly superior compared to NT-proBNP in HFpEF based on the qualitative analysis of AUC.

On the quantitative assessment, our study also confirmed the overall good diagnostic performance of GDF-15 based on pooled AUC. However, population-based disparities may have influenced our quantitative findings, particularly in the sensitivity analysis, in which diminished heterogeneity was observed when we omitted Santhanakrishnan et al. (2012), who conducted a study in Singapore. Importantly, the remaining three studies were conducted in Germany, where European cohorts may exhibit more distinct HFpEF vs. non-HFpEF characteristics (Ingimarsdóttir et al., 2025), making GDF-15 a stronger discriminator with a higher AUC. These findings suggest that geographic and racial differences may influence the overall diagnostic accuracy of GDF-15 in HFpEF, emphasising the need for a population-based cut-off, further validating its reliability in diverse cohorts.

AF is one of the main comorbidities found in HFpEF, with a prevalence ranging from 40% to 60% (Fauchier, Bisson & Bodin, 2023). Our study showed that GDF-15 was higher in the population with AF without HF compared to the HFpEF population. AF is linked to systemic inflammation, oxidative stress, and structural alterations in the heart, perhaps resulting in increased levels of GDF-15 even in populations without HF. Moreover, AF frequently coexists with comorbidities such as hypertension, diabetes, and obesity, which exacerbate inflammatory and oxidative stress pathways, elevating GDF-15 levels (Sartipy et al., 2017; Zafrir et al., 2018). This particular mechanism may influence our result, as the comparator group with AF comorbidities may have had inherently high GDF-15 levels. This effect was observed in our sensitivity analysis, where omitting the studies by Oyama et al. (2023), Aulin et al. (2022) resulted in a narrowed difference in GDF-15 levels between HFpEF and non-HFpEF groups. Therefore, further studies are needed to determine the optimal cut-off to use GDF-15 in diagnosing HFpEF in the setting of the AF population.

GDF-15 is an attractive diagnostic marker for HF patients across their ejection fraction and could play an important role in diagnosis in specific HF patients with attenuated levels of BNP or NT-pro BNP. One such important group are those with accompanying comorbidities of obesity and metabolic syndrome (Madamanchi et al., 2014; Kozhuharov et al., 2019). Recent studies have shown that BNP and NT-pro BNP levels are compromised in obese patients. In the GUIDE-IT trial involving 873 HFrEF patients, NT pro-BNP levels were 59.0% (95% CI [39.5–83.5%]) lower in obese patients (≤1,000 pg/mL). Yet despite this, those with higher BMI were associated with a greater risk of adverse cardiovascular events, regardless of low or high NT pro-BNP levels (Felker et al., 2017). This finding was further confirmed in the Asian population through the Suita study, showing its cross-cultural validity (Sugisawa et al., 2010). The strongest predictors of low BNP levels found were high BMI, younger age, higher LVEF, and lower creatinine (Bachmann et al., 2021). Hence, in obese patients with HFpEF, diagnosis through the use of conventional cardiac biomarkers would be a unique challenge. Our study demonstrated that GDF-15 could be a potential biomarker for HFpEF diagnosis in this subgroup of patients, as GDF-15 were significantly higher in the obese HFpEF population than in the obese population with normal LV function. However, further studies are needed to verify this result, as only one of the studies included in our analysis contains this subgroup of population.

The incidence of HFpEF, in particular is still rising even while the overall prevalence of HF seems to be constant or even declining (Tsao et al., 2018). According to four community-based cohorts, the incidence rate of HFpEF is estimated to be 27 cases per 10,000 person-years, and it is anticipated that its prevalence is going to surpass that of HFrEF. HFpEF also has a comparably poor clinical outcome with HFrEF (Bhambhani et al., 2018). Our study has shown that elevated GDF-15 is significantly associated with all-cause mortality and HF hospitalisation, making it a valuable prognostic marker.

This meta-analysis evaluates both the diagnostic and prognostic value of GDF-15 in the HFpEF population. It involves a wide range of studies conducted in different countries and includes HFpEF patients in both the acute and chronic clinical settings. The majority of studies included, except one, were all of good quality and provided consistent results. However, several limitations should also be noted. There was apparent heterogeneity across all studies, which may have impacted the study results. One of the major sources of heterogeneity arises from the utilisation of various GDF-15 assay kits across studies, including research-grade Enzyme-linked immunosorbent assay (ELISA) kits (e.g., Biovendor, R&D Systems, Millipore) and the in-vitro diagnostic grade Roche Diagnostics Elecsys electrochemiluminescence immunoassay. Research-grade ELISA kits exhibit a coefficient of variation of roughly 10% to 20%, while the Roche Elecsys assay demonstrates a coefficient of variation of less than 4% (Karusheva et al., 2022). This disparity may result in variations in assay precision and measurement accuracy. Moreover, differences in HFpEF comorbidities and race among the study populations may have further contributed to study heterogeneity. However, due to limited data, we were unable to perform subgroup analysis or meta-regression. Future studies should aim to standardise GDF-15 measurement techniques to improve data consistency and reliability in meta-analysis. The second limitation is that only limited studies were included in the prognostic meta-analysis; therefore, the interpretation of these results requires caution. Additional research involving larger sample sizes is required to enhance the evidence in this domain.

Although our study has demonstrated the significance of GDF-15 in HFpEF diagnosis and prognosis, several challenges still need to be addressed before it can be widely implemented in clinical practice. First, an optimal cut-off value of GDF-15 for risk stratification still needs to be elucidated. Second, the effects of comorbidity, such as AF, on the diagnostic value of GDF-15 need to be considered. Third, our study was unable to assess the relationship between GDF-15 and specific hemodynamic phenotypes of HFpEF, as distinct profiles phenotype may present with differing levels of GDF-15, potentially reflecting varying underlying pathophysiology and severity. Therefore, large multi-centred prospective studies are still required for these questions. In the future, based on its extensive pathophysiological mechanism, GDF-15 also carries potential for therapeutic applications, and studies are currently ongoing on its effectiveness against cardiometabolic disease, non-alcoholic fatty liver disease, and obesity (Wang et al., 2021).

Conclusions

GDF-15 could serve as a promising biomarker in HFpEF, with the overall diagnostic performance appearing to be slightly superior compared to conventional biomarkers and also demonstrating significant prognostic potential in HFpEF. Higher levels of GDF-15 are linked to increased risk of all-cause mortality and HF hospitalisation.

Supplemental Information

Supplemental Information 1 PRISMA checklist.

Supplemental Information 2 Supplemental Tables.

Supplemental Information 3 Audience Intended For.

Additional Information and Declarations

Competing Interests

The authors declare that they have no competing interests.

Author Contributions

Iwan Dakota conceived and designed the experiments, analyzed the data, prepared figures and/or tables, authored or reviewed drafts of the article, and approved the final draft.

Matthew Aldo Wijayanto conceived and designed the experiments, performed the experiments, analyzed the data, prepared figures and/or tables, authored or reviewed drafts of the article, and approved the final draft.

Annisa Salsabilla Dwi Nugrahani analyzed the data, authored or reviewed drafts of the article, and approved the final draft.

Angela Felicia Sunjaya performed the experiments, authored or reviewed drafts of the article, and approved the final draft.

Shela Rachmayanti performed the experiments, authored or reviewed drafts of the article, and approved the final draft.

Enny Yuliana Indah analyzed the data, authored or reviewed drafts of the article, and approved the final draft.

Natasya Naomi analyzed the data, authored or reviewed drafts of the article, and approved the final draft.

Wilbert Huang analyzed the data, authored or reviewed drafts of the article, and approved the final draft.

Christopher Daniel Tristan analyzed the data, prepared figures and/or tables, authored or reviewed drafts of the article, and approved the final draft.

Mira Fauziah conceived and designed the experiments, authored or reviewed drafts of the article, and approved the final draft.

Hary Sakti Muliawan conceived and designed the experiments, authored or reviewed drafts of the article, and approved the final draft.

Bambang Budi Siswanto conceived and designed the experiments, authored or reviewed drafts of the article, and approved the final draft.

Data Availability

The following information was supplied regarding data availability:

Raw data was not generated in this systematic review/meta-analysis.

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
