# Peer review of "Diagnostic and prognostic implications of growth differentiation factor 15 in heart failure with preserved ejection fraction: a systematic review and meta-analysis"

_PeerJ, doi:10.7717/peerj.20168_

## Round 0.1 · original submission · Major Revisions

**Language Note:** The review process has identified that the English language must be improved. PeerJ can provide language editing services - please contact us at [email protected] for pricing (be sure to provide your manuscript number and title). Alternatively, you should make your own arrangements to improve the language quality and provide details in your response letter. – PeerJ Staff

Reviewer 1 ·

Basic reporting

This study aimed to assess the diagnostic and prognostic significance of GDF-15 in HFpEF.
12 observational studies were included in this systematic review and meta-analysis.
GDF-15 levels were consistently elevated in HFpEF patients in comparison to control groups.

On qualitative analysis, the overall diagnostic performance of GDF-15 in HFpEF appeared to be slightly superior compared to conventional biomarkers. Moreover, GDF-15 proved to be a valuable prognostic biomarker for evaluating all-cause mortality and HF hospitalisation.

Accordingly, the overall diagnostic performance of GDF-15 appeared to be slightly superior compared to conventional biomarkers and also demonstrated significant prognostic potential in HFpEF.

Experimental design

The experimental design is well presented.
The research question is well defined.
The methods are complete.
The statistical analysis is adequate.

Validity of the findings

The authors' findings are original.
The authors demonstrated and validated the incremental diagnostic and prognostic role of GDF-15 in HFpEF patients.
The conclusions are supported by the results of the study.

Additional comments

I have one suggestion for the authors.

At the end of the Discussion section or within the Limitations section, the authors must address the relationship between GDF-15 and the hemodynamic phenotypes of HFpEF. Recent evidence indicates that HFpEF prognosis is closely linked to specific hemodynamic profiles encountered in clinical practice: (1) warm (no signs or symptoms of hypoperfusion) and dry (no signs or symptoms of congestion); (2) warm (no hypoperfusion) and wet (presence of any signs or symptoms of congestion); (3) cold (presence of any signs or symptoms of hypoperfusion) and wet (congestion) and; (4) cold (hypoperfusion) and dry (no congestion) (PMID: 27206819). Given the particularly poor prognosis associated with the “cold and dry” phenotype (PMID: 34988931), it is likely that GDF-15 levels are significantly elevated in this subgroup, often associated with septic conditions. This potential relationship should be discussed, and the need for further studies evaluating GDF-15 levels across these phenotypes must be emphasized.

Reviewer 2 ·

Basic reporting

The structure conforms to the PeerJ standards. The figures are relevant and well labelled. My concerns with the introduction and background are that text lines 55 to 67 are unnecessary and do not enhance the paper's credibility. The article, however, was adequately referenced.

The authors were also able to demonstrate both the diagnostic and prognostic roles of GDF-15 in patients with HFpEF; however, larger studies would be required to determine if it can influence practice guidelines.

Experimental design

The abstract can be improved. My suggestions:

1. Background: GDF-15 is a biomarker and should not be termed an instrument for uniformity.

Methods: Mention how the outcomes were pooled together with the confidence intervals. Cochrane advises that we use the random effects model.

Results: The pooled effects of patients with HFpEF and the controls should be stated, and then those of patients with atrial fibrillation should be added.

How many patients with HFpEF and controls were included? Just mention them in the opening statement of the results section.

Conclusion: This can be improved upon.

2. Body
Introduction: My comment is as stated above.
Line 106 would require a reference.
Line 125: Evaluators should be changed to authors
Line 160: Was the number 11818 a mistake? Was there a study that included this number of participants?
Line 180/181: Make it more straightforward, likely with the repositioning of the word exclusion.
Lines 201-204: Only present your findings as you don't need to compare with other studies here.
Line 214: It appears that there were no controls, as all 5,696 patients had HFpEF.
Clearly state the number of participants with HFpEF or the number of controls, if applicable. You don't have to mention the two.
Lines 223-226: You can mention some of these alterations that relate to GDF-15.
Lines 235-242: Are these confounding factors related only to patients with HFpEF or across the continuum of EF?
Line 246: Can you explain why, after the leave-one-out sensitivity analysis, the heterogeneity remained high, but you said it dropped?
Lines 254-256: Why did you include studies with AF without HFpEF?
Lines 286-290: I am unsure whether the statement that you were the first to carry out this meta-analysis is necessary.

Validity of the findings

Forest plots: You can change the label Experimental to HFpEF
Please indicate below the forest plot whether it favours HFpEF or the control.

Table 1: You can make it more aesthetic by reducing the number of variables eg by removing columns 9 and 10.

You may also wish to abbreviate prospective as you did for RCTs

Move site together with Country column (Site, Country: eg 2, USA, Germany)

Why are Izumiya et al and Mendez-Fernandez et al not appearing in any of your forest plots?

If you did not pool any of their outcomes, then they should not be among the included studies.

Additional comments

The grammar should be improved.

---

## Round 0.2 · accepted · Accept

Thank you for addressing the reviewers' comments.

Reviewer 1 ·

Basic reporting

The authors' findings are original and very interesting for clinical cardiologists.

Experimental design

-

Validity of the findings

-

Additional comments

After the following sentence: "Third, our study was unable to assess the relationship between GDF-15 and specific hemodynamic phenotypes of HFpEF, as distinct phenotypes may present with differing levels of GDF-15, potentially reflecting varying underlying pathophysiology and severity", the authors could provide more references, for example, PMID: 27206819

Reviewer 2 ·

Basic reporting

Ok

Experimental design

Ok

Validity of the findings

Ok

Additional comments

All forest plots must have favour GDF-15 or favour placebo below it.
PRISMA flow diagram: On the part included: Studies included in qualitative review: 12
Studies included in quantitative review: 11